# Spatial Distribution, Potential Sources, and Health Risk of Polycyclic Aromatic Hydrocarbons (PAHs) in the Surface Soils under Different Land-Use Covers of Shanxi Province, North China

**DOI:** 10.3390/ijerph191911949

**Published:** 2022-09-21

**Authors:** Li Ji, Wenwen Li, Yuan Li, Qiusheng He, Yonghong Bi, Minghua Zhang, Guixiang Zhang, Xinming Wang

**Affiliations:** 1School of Environment and Resources, Taiyuan University of Science and Technology, Taiyuan 030024, China; 2State Key Laboratory of Freshwater Ecology and Biotechnology, Institute of Hydrobiology, Chinese Academy of Sciences, Wuhan 430072, China; 3State Key Laboratory of Organic Geochemistry, Guangzhou Institute of Geochemistry, Chinese Academy of Sciences, Guangzhou 510640, China

**Keywords:** carcinogenic risk, GIS, land-use management, PAHs, spatial pattern

## Abstract

Polycyclic aromatic hydrocarbons (PAHs) are widespread in the environment and pose a serious threat to the soil ecosystem. In order to better understand the health risks for residents exposed to PAH-contaminated soil, 173 surface soil samples were collected in Shanxi Province, China, to detect the levels of 16 priority PAHs. The spatial distribution patterns of PAHs were explored using interpolation and spatial clustering analysis, and the probable sources of soil PAHs were identified for different land-use covers. The results indicate that the soil Σ16 PAH concentration ranged from 22.12 to 1337.82 ng g^−^^1^, with a mean of 224.21 ng g^−^^1^. The soils were weakly to moderately contaminated by high molecular weight PAHs (3–5 ring) and the Taiyuan–Linfen Basin was the most polluted areas. In addition, the concentration of soil PAHs on construction land was higher than that on other land-use covers. Key sources of soil PAHs were related to industrial activities dominated by coal burning, coking, and heavy traffic. Based on the exposure risk assessment of PAHs, more than 10% of the area was revealed to be likely to suffer from high carcinogenic risks for children. The study maps the high-risk distribution of soil PAHs in Shanxi Province and provides PAH pollution reduction strategies for policy makers to prevent adverse health risks to residents.

## 1. Introduction

Polycyclic aromatic hydrocarbons (PAHs) are contaminants abundant in the environment [1]. They originate from either natural sources (e.g., biosynthesis process, natural fires, and volcanic eruptions) or anthropogenic activities (e.g., the incomplete combustion of fossil fuels, coke production, gasoline, lignin, sediment, and diesel from fossil fuels) [2,3]. Both the ecosystem and humans are seriously threatened by the potential carcinogenic and mutagenic risk of PAHs [4]. PAH exposure on humans is associated with a variety of diseases, such as lung cancer and neural tube defects [5]. For example, benzo[a] pyrene (Bap), a typical PAH, has been reported as a carcinogen with the ability to promote cancer in bladders, lungs, prostates, breasts, and other organs based on epidemiological studies [6]. Thus, the pollution of PAHs has attracted widespread concern from the scientific community.

The soil ecosystem, one of the most threatened and heavily used natural systems globally, is generally considered to be a major PAH sink in the environment [7]. Atmospheric PAHs are continuously deposited into surface soils by wet or dry deposition processes and are eventually bound to soil particles [8]. Soil PAHs are more resistant to degradation due to the stable organic matter fractions. In addition, they are easily adsorbed by particles and migrate to other locations due to their hydrophobicity [9,10]. According to the Bulletin of Chinese Soil Survey, 16.10% of soil in China is contaminated by pollutants such as heavy metals and organic matter, which exceeds the standard rate of PAHs by 1.40% [11].

The distribution of soil PAHs in the environment has been significantly influenced by regional economic development, energy composition, and population density [12]. PAHs are released into the soil primarily through industrial activities such as the burning of fossil fuels for electricity generation, vehicle emissions, petroleum refining, chemical manufacturing, and industrial processing [13]. For example, [14] reported concentrations of 16 PAHs (∑16 PAHs) in agricultural soil close to a chemical plant ranging from 250.49 to 9387.26 ng g^−^^1^ with a mean of 2781.42 ng g^−^^1^. PAHs in relatively remote, sparsely populated areas far away from industrial activities have also been detected due to their environmental persistence and long-distance atmospheric transport [15]. In particular, PAHs were detected in the soils of tropical areas in the absence of industrial point sources, and even in polar regions [16]. Moreover, the emitted PAHs in China were determined to be approximately 32,720 tons due to the incomplete combustion of fuel in 2016, with the contribution order of the emission sources as follows: biomass combustion > residential coal combustion > vehicle > coke production > refine oil > power plant > natural gas combustion [17]. Coal usage is one of the principal sources of PAHs in China, and is a factor driving the higher PAH soil levels in northern China (1467 ng g^−^^1^ in the northeast and 911 ng g^−^^1^ in the north) compared to the eastern (737 ng g^−^^1^), southern (349 ng g^−^^1^), and western (209 ng g^−^^1^) regions [18].

Shanxi Province is a typical heavy industry province in northern China and is famous for its abundant coal resources. The Carboniferous–Permian coal and coking coal resources in this coal-accumulation zone account for one-quarter and three-quarters of the total coal resources in China, respectively [19]. As one of the largest energy and industrial bases in China, Shanxi Province covers just one-sixtieth of China’s territory, yet its coke production and power generation accounts for two-fifths and one-seventeenth, respectively [19]. Although the soils of Shanxi have been reported as contaminated with extensive and severe PAHs [14,20], the spatial clustering and regional risk distribution of soil PAHs proves to be a complicated task [21,22]. In addition, research on the differences in the sources and distribution of soil PAHs under different land-use covers is limited. In order to overcome these limitations, the current study aims to: (1) understand the spatial distribution pattern of soil PAHs in Shanxi Province; (2) identify the sources of soil PAHs under different land-use covers; and (3) assess the health risk of soil PAHs. This study can help to understand the impacts of anthropogenic activities on soil PAHs in typical industrial areas and provide useful information for policy -makers to effectively control pollution.

## 2. Materials and Methods

### 2.1. Study Area

Shanxi Province (110°14′ N–114°33′ N, 34°34′ E–40°44′ E), one of the largest coking coal production bases worldwide, is situated in the coal-accumulation zone of North China (Figure 1a). It is located in the eastern Loess Plateau with a diverse topography, including mountains, hills, basins, and plateaus. Shanxi is a typical semi-arid area with four distinct seasons, and the annual average precipitation ranges from 358 to 621 mm.

### 2.2. Sample Collection

A total of 173 surface soil samples from Shanxi Province were collected in 2020: 34 from grassland, 14 from forest, 89 from farmland, and 36 from construction land (including urban, suburban, and industrial land) (Figure 1b). Data on land-use cover map were collected from Resource Environment Data Cloud Platform (http://www.resdc.cn/ accessed on 1 January 2022) and were accurate to 1 km. For each sampling site, 5 subsamples (each 1 kg) were taken from the same area (at a depth of 0–20 cm in a 100 m^2^ square area) with a stainless steel shovel, and then bulked together to form one composite sample. A quarter of the composite sample was collected and stored in a polyethylene bag. Stones and residual roots in the soil samples were removed and freeze-dried for 48 h. Soil samples were sieved into 70 mesh particles prior to analysis. The locations of the samples were recorded using the Global Position System (GPS, GISIBAO G138BD).

### 2.3. Chemical Analysis and Quality Control

A 10.0 g soil sample was treated with 10.0 g of anhydrous sodium sulfate (1/1, v/v) and 2.0 g of activated copper using an automated solvent extractor (Dionex ASE 300, Sunnyvale, CA, USA). The mixture was spiked with 20.0 mL n-hexane/dichloromethane (1/1, v/v), which was used as an extraction solvent. The PAHs (20 ppm, d8-NAP, d10-ACE, d10-ANT, d12-CHR, and d12-Perelyne) were employed as surrogates. Following the addition of 30.0 mL of n-hexane/acetone, the mixture was extracted with ultrasound for 10 min [23]. The extracts were then concentrated to 2.0 mL using a rotary evaporator, loaded onto a silica gel aluminum oxide chromatographic column for cleanup, and further concentrated to 1.0 mL in a nitrogen stream prior to analysis [14]. The eluent was solvent-changed and concentrated to 1.0 mL before analysis. All PAHs were measured by gas chromatography -mass spectrometry (Shimadzu GC-MS 2010 plus, Japan) using an HP-5MS capillary column (length: 30 m, inner diameter: 0.32 mm, film thickness 0.25 mm). The analytical procedure was performed according to the US EPA Method 8270D, 3550B. The standard samples of the PAHs were analyzed using the peak area of an external reference (Supelco Co., Sigma-Aldrich Corporation, St Louis, MO, USA).

The 16 priority PAHs were as follows: naphthalene (Nap); acenaphthylene (Acy); acenaphthene (Ace); fluorene (Flu); phenanthrene (Phe); anthracene (Ant); fluoranthene (Fla); pyrene (Pyr); benz[a] anthracene (Baa); chrysene (Chr); benzo[b] fluoranthene (Bbf); benzo[k] fluoranthene (Bkf); benzo[a] pyrene (Bap); indeno[1,2,3-cd] pyrene (Inp); dibenzo[ah] anthracene (Daa); and benzo[ghi] perylene (Bgp).

The relative standard deviation of the standard curve was less than 20% and a linear quantitative equation was obtained with an r^2^ > 0.99. The method detection limits ranged from 10.0 to 15.0 ng g^−^^1^ and the recoveries were determined within the range of 48.14–123.88%, including d8-Nap (48.14–82.68%), d10-Ace (57.78–86.29%), d10-Phe (60.61–104.13%), d12-Chr (66.91–115.70%), and d12-Pyr (73.45–123.88%).

### 2.4. Source Apportionment

Positive matrix factorization (PMF) calculations were conducted based on the US EPA (Environmental Protection Agency) PMF 5.0 [24,25]. The PAH pollution in Shanxi soil was grouped into three key sources, namely, coal and biomass, coke, and fossil oil and traffic.

### 2.5. Carcinogenic Risk Assessment

BaP toxic equivalent concentration (BaP*_eq_*) was applied to assess the health risk of PAHs and was calculated by multiplying the concentration of individual PAHs and the corresponding toxicity equivalent factor (TEF) value [26,27]. Carcinogenic risk (CR) of soil PAHs was assessed through three pathways including accidental ingestion, skin contact, and respiratory ingestion for people of different ages (including children, youths, and adults). Risk assessment methods and specific parameters are provided in the Appendix A. The inverse distance weighting (IDW) method was used to map the risk of soil PAHs. The US EPA guidelines are as follows: no CR for risk levels less than 10^−6^; potential CR for risk levels between 10^−6^ and 10^−4^; and high CR for risk levels greater than 10^−4^ [28]. In this study, four categorical risk levels were created, namely very high (CR > 4 × 10^−5^), high (2 × 10^−5^ < CR < 4 × 10^−5^), moderate (10^−5^ < CR < 2 × 10^−5^), and low (CR < 10^−5^).

### 2.6. Statistical and Geostatistical Analysis

Multivariate statistical techniques including the creation of histograms, stacked histograms, and pie charts were run on SigmaPlot (Systat Software, 14.0.lnk, USA). Variations between different groups were determined using ANOVA with a significant set of *p* < 0.01. Box pots were drawn with R (R 4.1.3, AT&T Bell Laboratories) [29]. The Moran index and bivariate local indicators of spatial association (LISA) can effectively identify the spatial clustering and regional risk distributions of environmental pollutants [30,31]. The combination of pollutant source identification under different land-use covers and risk assessment models can comprehensively provide key information for pollution control and strategy development. All the spatial visualization maps were mapped using ArcGIS Desktop 10.2 (Esri, Redlands, CA, USA). Spatial autocorrelation (Global Moran’s Ⅰ, between −1 and 1) was employed to analyze the spatial characteristics of soil samples [32]. Cluster and outlier analyses (Anselin Local Moran’s Ⅰ) were performed for the pollutants with a detected spatial correlation. As not all PAHs exhibited a significant spatial autocorrelation, the inverse distance weighting method (IDW interpolation) from the ArcGIS geostatistical analyst toolbox was used for the spatial prediction of pollutants [33], with an interpolation power of 1, and the maximum and minimum neighbors set to 15 and 10, respectively.

## 3. Results

### 3.1. Concentration and Compositional Profiles of Soil PAHs

Table 1 reports the concentrations of individual PAHs, the PAHs with different ring numbers, and the sum of the 16 PAHs (∑16 PAHs) in the soil from Shanxi Province. The total concentrations of ∑16 PAHs ranged from 22.12 to 1337.82 ng g^−1^, with a mean of 224.21 ng g^−1^. Phe and Bbf were observed to have the highest concentrations of the 16 PAHs and seven carcinogenic PAHs (based on the US EPA), at 32.92 ng g^−1^ and 29.89 ng g^−1^, respectively. The concentration of Phe was 20 times higher than that of Acy, which exhibited the lowest concentration (1.61 ng g^−1^).

Figure 2a presents the PAH profiles in all soil samples. The major PAHs were identified as Phe, Bbf, and Fla, accounting for 13.64%, 13.45%, and 12.48% of the total PAHs, respectively. The least contribution of PAHs was from Acy (0.40%) followed by Ace (0.44%). Figure 2b depicts the PAH proportions with different ring numbers. The 4-ring PAHs exhibited the highest percentage (37.58%) in the soil samples, followed by the 5-ring (23.94%), 3-ring (19.80%), 6-ring (13.56%), and 2-ring (5.12%) PAHs. High molecular weight PAHs (3–5 ring) accounted for 81.32% of the total PAHs.

### 3.2. Spatial Distribution Pattern of Soil PAHs

Figure 3 presents the predicted spatial distribution pattern of the soil ∑16 PAHs and PAHs with different ring numbers. The 4-ring, 5-ring, 6-ring, and ∑16 PAHs exhibit a similar spatial distribution pattern, with distribution relatively higher in the center and west, and lower in the north, south, and east of Shanxi Province. High concentrations of 2-ring and 3-ring PAHs were observed in Linfen Basin (Figure 3a,b). Note that high concentrations of PAHs were mainly concentrated in Linfen Basin and Taiyuan Basin.

The spatial autocorrelation analysis of the soil PAHs reveals a significant cluster distribution for the 2-, 4-, and 6-ring PAHs (Table 2). Higher and lower concentrations of the 2-ring PAHs were mainly concentrated in the southeast of Lvliang and the east of Taiyuan, respectively. Lower concentrations of the 4- and 6-ring PAHs were generally clustered in the southern border of Shanxi Province. The soil 4- and 6-ring PAHs in the east of Taiyuan demonstrated a mixed clustering trend, including “High–High” Cluster and “Low–High” Outlier (Figure 4).

### 3.3. Diversity of Soil PAHs under Different Land-Use Covers

Construction land exhibited the highest average level of ∑16 PAHs (226.18 ng g^−^^1^), followed by farmland (138.93 ng g^−^^1^). The lowest level was determined for grassland and forest (100 ng g^−^^1^) (Figure 5). The average levels of 2- to 6-ring PAHs were also maximized on construction land (11.50, 41.88, 84.09, 56.45, and 33.66 ng g^−^^1^, respectively), followed by farmland, with the exception of 2-ring PAHs (farmland: 36.39 ng g^−^^1^; forest: 37.95 ng g^−^^1^) (Figure 5).

The proportion of 4-ring PAHs was the highest, while that of the 2-ring PAHs was the lowest under different land-use covers (Figure 6a). More specifically, Phe (32.17 ng g^−^^1^), Bbf (26.32 ng g^−^^1^), and Flt (25.60 ng g^−^^1^) concentrations were maximized under farmland. A similar trend was also observed in grassland (Phe 24.88, Bbf 18.45, and Flt 13.99 ng g^−^^1^) (Figure 6b,e). Peak concentrations of Bbf (35.34 ng g^−^^1^), Phe (32.36 ng g^−^^1^), and Flt (28.49 ng g^−^^1^) were determined under forest, closely followed by construction land (Bbf 45.65, Phe 42.14, and Flt 38.47 ng g^−^^1^) (Figure 6c,d).

### 3.4. Source of Contamination

The relative contributions of the three identified sources to the total PAHs in the soil samples were 43.1%, 35.6%, and 21.3% for coal and biomass combustion, coke, and fossil oil and traffic, respectively (Figure 7). The contribution of pollution sources varied with the land-use cover type. Coal and biomass combustion contributed to 37.1%, 35.7%, 31.3%, and 26.5% of the total sources in grassland, farmland, forest, and construction land soils, respectively. The contribution of PAH sources also varied with the land-use covers. The coke industry contributed to 36.3%, 30.7%, 27.9%, and 16.4% of PAH sources in farmland, grassland, forest, and construction land soils, respectively, while fossil oil and traffic contributed 57.1%, 40.8%, 32.2%, and 28.0%, respectively (Figure 7).

### 3.5. Carcinogenic Risk

The BaP toxic equivalent concentration (BaP*_eq_*) of 16 PAHs in the soil of Shanxi Province ranged from 2.20 to 974.89 ng g^−^^1^, with an average value of 30.81 ng g^−^^1^ (Appendix A). The BaP*_eq_* of the seven carcinogenic PAHs ranged from 0.96 to 180.40 ng g^−^^1^ with an average of 20.57 ng g^−^^1^, accounting for 97.53% of Σ16 PAHs BaP*_eq_*. The seven carcinogenic PAHs contributed the most to the carcinogenic risk (Appendix A). BaP (11.25 ng g^−^^1^) alone accounted for about 52.02% of the total BaP*_eq_*. A total of 10.78%, 60.20%, and 1.39% of children, youths, and adults, were determined to be at risk, respectively, and 0.17%, 0.07%, and 0.02% at very high risk, respectively (Figure 8). The high-risk areas of soil PAHs for children and youths were mainly distributed in Taiyuan Basin and Xinzhou Basin in central Shanxi Province (Figure 8).

## 4. Discussion

### 4.1. Concentration, Composition Profiles, and Spatial Distribution Pattern of Soil PAHs in Shanxi Province

The European criteria of soil contamination classifies the contamination into four grades: ∑16 PAHs < 200 ng g*^−^*^1^, no contamination; 200–600 ng g*^−^*^1^, weak contamination; 600–1000 ng g*^−^*^1^, moderate contamination; >1000 ng g*^−^*^1^, heavy contamination [34]. Based on this, the soil samples in this study were weakly contaminated for ∑16 PAHs (average value 224.21 ng g*^−^*^1^). The maximum value of ∑16 PAHs (1337.82 ng g^−1^) was 1.34 times higher than the heavily contaminated level (1000 ng g^−1^). The mean concentration of ∑16 PAHs was lower than previously reported in North China (911 ng g^−1^ [18]; 336 ng g^−1^ [35]) and Shanxi Province (3456 ng g^−1^ [36]; 2780 ng g^−1^ [14]; 600 ng g^−1^ [27]). This may be attributed to the scattered locations of the sampling sites with a variety of soils in good condition (e.g., forest and grassland soils). The concentration of ∑16 PAHs determined in this study is also lower than that in economically developed provinces, including Shanghai (3290 ng g^−1^) [37], Guangdong (1503 ng g^−1^) [34] and Beijing (735 ng g^−1^) [38]. This is likely to be linked to the much larger and more direct industrial and vehicular emissions of these provinces.

High concentrations of PAHs were mainly located in Linfen Basin and Taiyuan Basin. The soil PAH distribution is a function of several factors, including the pollution source, emission density, and soil properties. Previous research has reported the soil PAHs in Shanxi Province to originate from mixed pyrogenic sources [27]. Coal is the largest mineral resource in Linfen with proven reserves of 39.8 billion tons, accounting for 14% of Shanxi Province. Linfen produces approximately 19.2 million tons of coke per year, accounting for 21.2% of Shanxi Province and generating 20.49 billion kwh of power [39]. Taiyuan Basin is the economic and political center of Shanxi Province, including Taiyuan, Lvliang, and Jinzhong City. Taiyuan Basin has many large-scale coking enterprises and coal production amounts to 10 million tons annually [40]. In 2021 Lvliang produced 145.75 million tons of raw coal, and 25 million tons of coke, and generated 33.03 billion kwh of electricity (http://www.lvliang.gov.cn accessed on 1 January 2022). Furthermore, in 2017, Jinzhong produced 72.89 million tons of raw coal, 11.61 million tons of coke, and generated 19.23 billion kwh of electricity (http://www.lvliang.gov.cn accessed on 1 January 2022). The large amount of coking and coal-burning power generation may be the most important factor of soil PAH contamination in these areas.

A large area of spatial clustering indicates different sources and spatial interaction [41]. High–high and low–high spatial clusters of the 4- and 6-ring PAHs were observed in the east of Taiyuan, implying that the high concentrations of soil PAHs may be affected by industrial emissions (coal burning, coking, and steel smelting) from surrounding Lvliang [42]. PAHs such as Bap can readily migrate into the atmosphere over long distances due to their hydrophobicity, to be deposited in surrounding areas [43]. In addition, as the capital of Shanxi Province, Taiyuan has 7243 km of open roads and a 104 km/100 km^2^ highway network density. Heavy traffic emissions are a key source of high PAH soil concentrations. The high–low and low–high spatial clusters of soil PAHs in Taiyuan may be potential problem areas, which need further attention [21]. Point source pollution is caused by mining industries; however, coking, smelting, and traffic are also critical, and the environmental quality management of urban areas in the vicinity of pollution sources cannot be ignored [44].

### 4.2. Sources and Diversity of Soil PAHs under Different Land-Use Covers

High molecular weight PAHs (3–5 ring, 81.32%) dominated the contaminated soils, suggesting the strong role of industrial activity and heavy traffic in PAH sources. High molecular weight PAHs (3–6 ring) are mainly generated by high-temperature combustion processes including vehicle combustion, coking, and industrial coal burning, while low molecular weight PAHs (<3 ring) are generally produced by moderate-to -low-temperature combustion processes including domestic coal and biomass burning [45]. PMF source identification further confirmed that industrial activities, dominated by coal combustion (coal: 43.1%; coke: 35.6%) were the main driving factors of soil PAH pollution in this study.

The highest concentration of ∑16 PAHs was observed for construction land, followed by farmland. This is consistent with the China Ministry of Environmental Protection, with the soil PAHs of farmland, forest, and grassland reported to exceed the standard by 19.4%, 10.0%, and 10.4%, respectively [11]. Intensive urbanization and industrialization increase PAHs in soil through human activities such as coal burning for electricity generation, coking, petroleum refining, industrial processing, vehicle combustion, and chemical manufacturing [46]. The high concentrations of Bbf, Phe, and Flt under construction land observed in this study also indicate industrial activity as a source of PAHs. PMF source analysis demonstrated the PAHs on construction land to mainly originate from fossil fuel and vehicle combustion. Therefore, industrial and traffic emissions are key areas in controlling PAH pollution for construction land. The power industry, transport industry, and tourism introduced by agricultural activities increase soil PAH levels [47]. In addition, the use of pesticides and the discharge of wastewater from agriculture severely affect the soil environment [48]. For example, higher levels of Phe in farmland (32.17 ng g^−1^) were observed in the current study, which may be caused by the spraying of pesticides containing Phe [49]. PMF source analysis indicated PAH pollution on farmland to be the comprehensive action of multiple sources.

In this study, forest soil PAHs were observed to mainly originate from traffic emissions (40.80%). Previous research has reported traffic emissions, coal burning, and residential biomass combustion as the three primary contributors to forest soil PAHs [50]. Moreover, the long-range transportation of PAHs via the atmosphere from construction land may also impact the PAH distribution in forest soils [51]. The uplift of air-masses in forests, as well as the increased precipitation amounts and the enhanced canopy scavenging effect may enhance the sink-function of forest soils for PAHs [52]. Forest ecosystems can play a vital role in scavenging anthropogenic PAHs [53], and the high richness of plant species on grassland may favor the biodegradation of PAHs [54]. Therefore, planning forests and grasslands around construction land will effectively reduce the impact of PAH pollution.

### 4.3. Exposure Risk of Soil PAHs

A total of 90 samples (52.02%) were determined to be at medium risk (100 ng g^−^^1^ < BaPeq < 500 ng g^−^^1^), and 4 samples (2.31%) were at high risk (500 ng g^−^^1^ < BaPeq < 1000 ng g^−^^1^). This implies that the majority of regions were threatened by a medium carcinogenic risk [55]. The BaP*_eq_* of seven carcinogenic PAHs ranged from 0.96 to 180.40 ng g^−^^1^, with an average of 20.57 ng g^−^^1^, accounting for 97.53% of Σ16 PAHs BaP*_eq_*. Among them, Bap presented the highest risk in this study. Bap is a 5-ring PAH and often originates from coal tar, the combustion of fossil fuel, cigarette smoke, and automobile exhausts, as well as industrial sewage from the coking, oil refining, asphalt, and plastic industries [56]. Bap can easily remain in the soil to subsequently pollute grain crops, fruits, and vegetables. Living for a long period of time in environments containing high concentrations of Bap can cause chronic poisoning and even lung cancer [57]. The coal industry, coking, and busy traffic in Shanxi Province are important factors for the formation of high concentrations of Bap [35]. The treatment of industrial waste and automobile exhaust is a highly effective measure in reducing the PAH exposure risk to residents [58].

All the soil samples from Shanxi Province were observed to pose carcinogenic risks from PAHs. Children are generally more active and consequently more exposed to the soil compared to youths and adults. The relatively lower weight of children is also a factor contributing to health risks [59]. PAHs can present a carcinogenic risk under the following forms (in descending order): dermal > ingestion > inhalation [60,61]. The predicted risk distribution results show that over 10% of the study area, mainly located in the south of Taiyuan and the north of Linfen, is likely to suffer from high carcinogenic risks for children. Residents in these areas require special protection. In addition, residents living on construction land suffer the highest exposure to risk. This is attributed to industrial activities and traffic emissions. This study mapped the high-risk distribution of soil PAHs for children, youths, and adults in Shanxi Province. Based on this, the following schemes were provided for policy -makers to determine effective and practical measures for the reduction in soil pollution to ensure residents’ health: (1) increase green areas; (2) use clean energy for industrial activities; and (3) develop new energy vehicles.

## 5. Conclusions

This study analyzed the contamination levels and explored the spatial patterns of PAHs in surface soils of a typical heavy industrial area, and subsequently identified the likely sources and assessed the potential health risks. The results indicated the soils to be weakly–moderately contaminated by high molecular weight PAHs (3–5 ring), with the Taiyuan–Linfen Basin the most polluted region. Industrial activities dominated by coal burning, coking, and heavy traffic were identified as the main sources of soil PAH. Construction land exhibited the highest soil PAH concentration among all land-use covers. Furthermore, the analysis revealed that over 10% of the study area is likely to suffer from high carcinogenic risks for children. 

The spatial distribution characteristics of pollutants can be effectively solved by means of spatial statistical analysis. The results urge the local government to weaken coal consumption and turn to clean energy to decrease PAH pollution and protect human health. In the future, more attention should be paid to the impact of long-term land-use change on the spatial distribution of soil pollutants.

## Figures and Tables

**Figure 1 ijerph-19-11949-f001:**
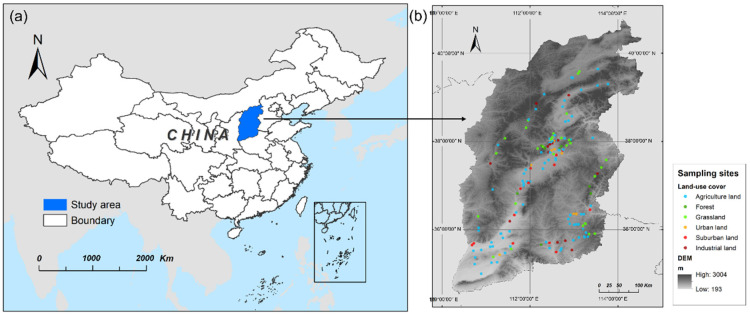
Location of (**a**) study area and (**b**) sampling sites under various land-use covers.

**Figure 2 ijerph-19-11949-f002:**
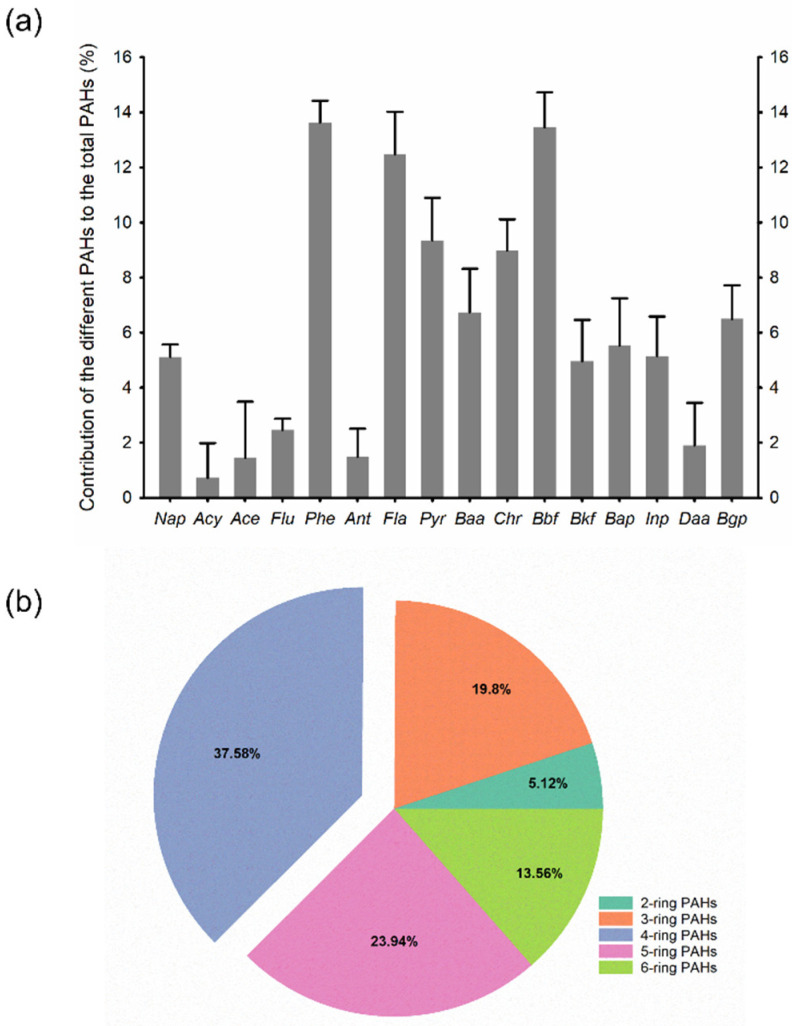
Proportion of individual PAHs (**a**) and PAHs with different ring numbers (**b**) in the soil samples. carcinogenic PAHs. Naphthalene (Nap); acenaphthylene (Acy); acenaphthene (Ace); fluorene (Flu); phenanthrene (Phe); anthracene (Ant); fluoranthene (Fla); pyrene (Pyr); benz[a] anthracene (Baa); chrysene (Chr); benzo[b] fluoranthene (Bbf); benzo[k] fluoranthene (Bkf); benzo[a] pyrene (Bap); indeno[1,2,3-cd] pyrene (Inp); dibenzo[ah]anthracene (Daa); and benzo[ghi] perylene (Bgp).

**Figure 3 ijerph-19-11949-f003:**
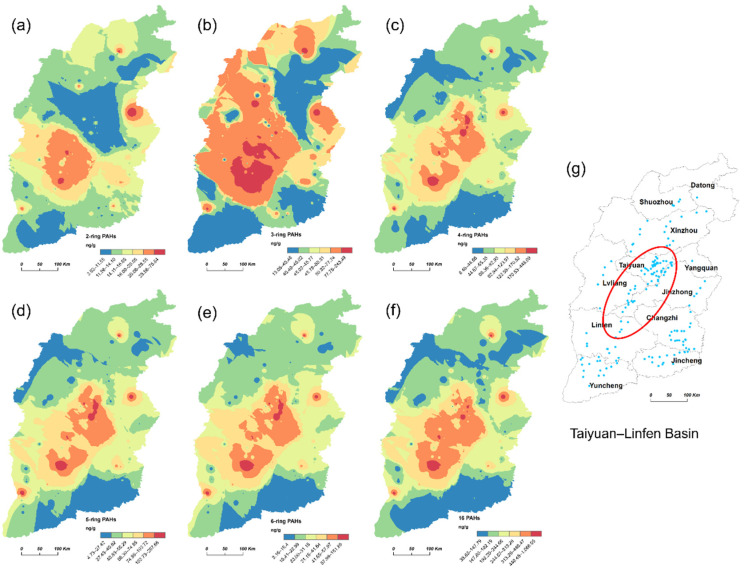
Spatial distribution pattern of soil ∑16 PAHs and PAHs with different ring numbers in Shanxi Province. (**a**) 2-ring PAHs; (**b**), 3-ring PAHs; (**c**) 4-ring PAHs; (**d**) 5-ring PAHs; (**e**) 6-ring PAHs; (**f**) ∑16 PAHs; (**g**) location of Taiyuan–Linfen Basin.

**Figure 4 ijerph-19-11949-f004:**
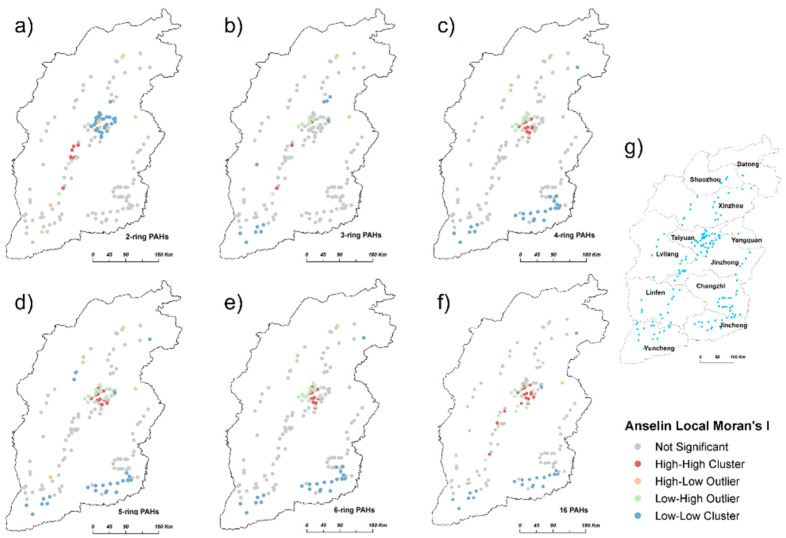
Clustering and outlier analysis of soil ∑16 PAHs and PAHs with different ring numbers in Shanxi Province, using the local indicators of spatial association method (LISA). (**a**) 2-ring PAHs; (**b**) 3-ring PAHs; (**c**) 4-ring PAHs; (**d**) 5-ring PAHs; (**e**) 6-ring PAHs; (**f**) ∑16 PAHs; (**g**) schematic diagram of administrative division of Shanxi Province.

**Figure 5 ijerph-19-11949-f005:**
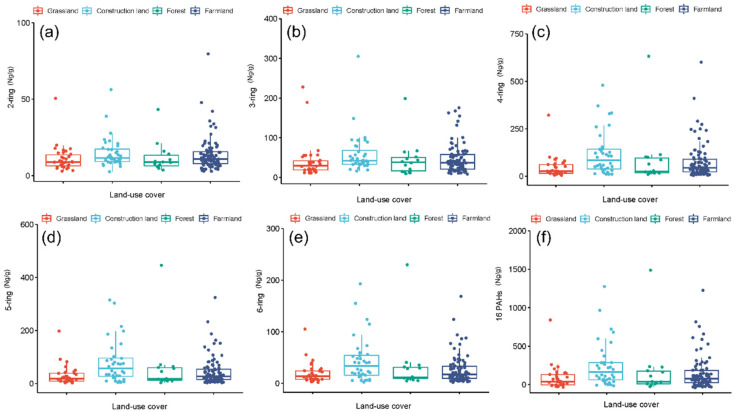
Boxplot of soil PAHs with different ring numbers (**a**–**e**) and ∑16 PAHs (**f**) under different land-use covers in Shanxi Province.

**Figure 6 ijerph-19-11949-f006:**
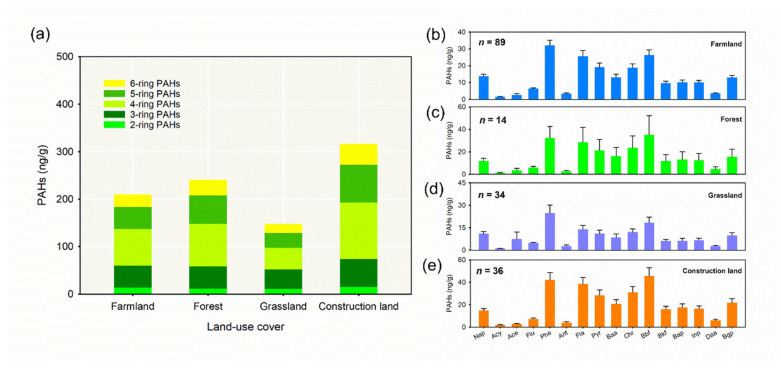
Stacked histogram of PAHs with different ring numbers (**a**) and histogram of 16 individual PAHs (**b**–**e**) under different land-use covers in Shanxi Province.

**Figure 7 ijerph-19-11949-f007:**
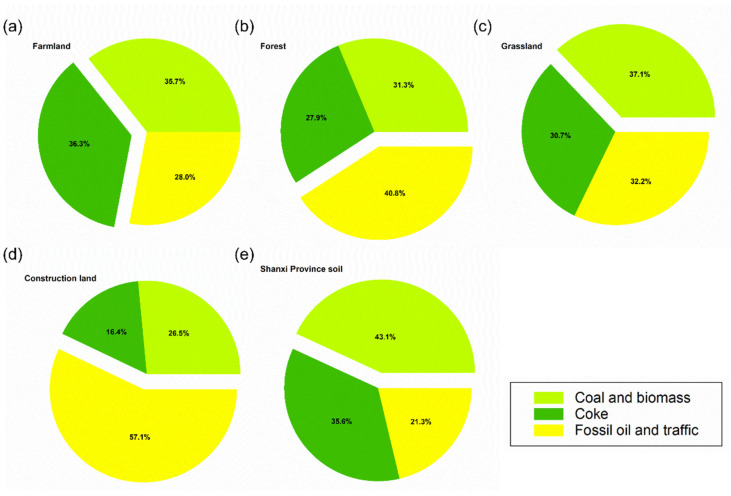
Source profiles and contribution of soil PAHs under farmland (**a**), forest (**b**), grassland (**c**), construction land (**d**), and all covers of land-use (**e**) obtained by PMF (positive matrix factorization).

**Figure 8 ijerph-19-11949-f008:**
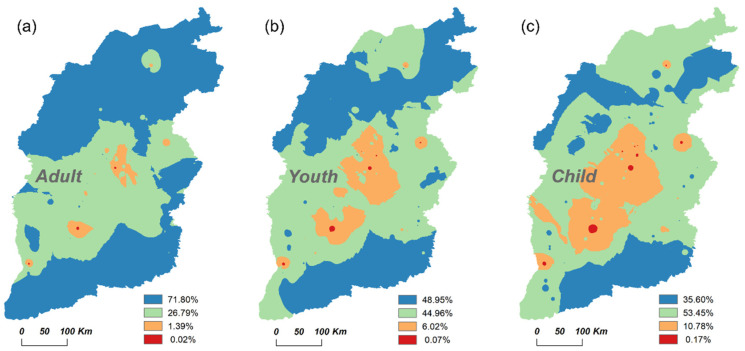
Spatial patterns and composition proportion of risk level of children (**a**), youths (**b**), and adults (**c**) in soil PAHs.

**Table 1 ijerph-19-11949-t001:** Concentrations of polycyclic aromatic hydrocarbons (PAHs) in the soil samples from Shanxi Province (ng g^−1^).

Compounds	Min (ng g^−^^1^)	Max (ng g^−^^1^)	Mean (ng g^−^^1^)	SD (ng g^−^^1^)	Compounds	Min (ng g^−^^1^)	Max (ng g^−^^1^)	Mean (ng g^−^^1^)	SD (ng g^−^^1^)
Nap	2.59	79.60	13.28	10.21	Bkf ^c^	0.30	72.85	10.60	13.11
Acy	0.09	10.92	1.61	1.45	Bap ^c^	0.21	89.79	11.38	15.69
Ace	0.21	167.11	3.76	13.56	Inp ^c^	0.29	71.62	11.13	13.36
Flu	1.11	31.07	6.33	4.19	Daa ^c^	0.18	25.25	4.05	4.68
Phe	5.03	235.34	32.92	32.12	Bgp	0.98	111.86	14.58	17.00
Ant	0.10	22.99	3.45	3.43	2-ring PAHs	2.59	79.60	13.28	10.21
Fla	1.51	221.37	26.48	33.61	3-ring PAHs	8.26	305.16	48.07	42.49
Pyr	1.08	160.87	19.85	25.14	4-ring PAHs	4.77	601.63	81.22	101.89
Baa ^c^	0.73	110.43	14.25	19.54	5-ring PAHs	2.33	324.73	51.87	65.32
Chr ^c^	1.20	148.79	20.63	25.07	6-ring PAHs	1.79	192.97	29.76	34.66
Bbf ^c^	1.06	181.07	29.89	36.88	∑16 PAHs	22.12	1337.82	224.21	243.99

Note: ^c^ carcinogenic PAHs. Naphthalene (Nap); acenaphthylene (Acy); acenaphthene (Ace); fluorene (Flu); phenanthrene (Phe); anthracene (Ant); fluoranthene (Fla); pyrene (Pyr); benz[a] anthracene (Baa); chrysene (Chr); benzo[b] fluoranthene (Bbf); benzo[k] fluoranthene (Bkf); benzo[a] pyrene (Bap); indeno[1,2,3-cd] pyrene (Inp); dibenzo[ah]anthracene (Daa); and benzo[ghi] perylene (Bgp).

**Table 2 ijerph-19-11949-t002:** Spatial autocorrelation analysis of the soil ∑16 PAHs and PAHs with different ring numbers.

PAHs	Moran’s I	*p*-Value	z-Score	Significance	Spatial Correlation
2-ring PAHs	0.09	0.06	1.85	Significant	Clustered
3-ring PAHs	0.03	0.37	0.90	Not significant	Random
4-ring PAHs	0.05	0.06	1.85	Significant	Clustered
5-ring PAHs	0.04	0.11	1.57	Not significant	Random
6-ring PAHs	0.04	0.09	1.68	Significant	Clustered
16 PAHs	0.04	0.15	1.44	Not significant	Random

## Data Availability

The datasets and materials used or analyzed during the current study are available from the corresponding author on reasonable request.

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
