# Peer review of "Spatial Distribution, Potential Sources, and Health Risk of Polycyclic Aromatic Hydrocarbons (PAHs) in the Surface Soils under Different Land-Use Covers of Shanxi Province, North China"

_ijerph, 2022, doi:10.3390/ijerph191911949_

Round 1

Reviewer 1 Report

Manuscript ijerph-1878341

The present manuscript shows the spatial occurrence, potential sources and health risk of 16 polycyclic aromatic hydrocarbons (PAHs) in soil surface samples collected from Shanxi Province (China), a relevant coking coal production and coal-accumulation area. It is a very complete study, since it offers quantitative data on PHA pollution in the zone (concentration and distribution) while also providing a vision of the potential risks to which the population is exposed. I find that the topic is of great interest and the contribution would provide valuable information on the behavior of these pollutants in the province, given the lack of knowledge on the issue. The manuscript is well written and the content is properly organized. The results are well presented and discussed with similar studies. The research fits within the Scopus of the journal.

In my opinion, the aforementioned merits are positive to propose this manuscript for acceptance for publication. Only minor revisions are required from my point of view. Details are listed below.

Section 2.3. According to the procedure, the extracts were concentrated to 1 mL under a stream of nitrogen, and then they were concentrated to 1-2 mL? It's not clear.

Line 190. Baf is not a studied PHA

Line 191. “The least dominant PAHs were Flu (0.40%) and Nap (0.44%)”. Are you referring to frequency or contribution? It does not correspond to Figure 2a. The one that contributes the least is Acy followed by Ace

Did the authors find similar studies on PAH removal? It would be interesting to compare the results with the main points / strategies for elimination or degradation of the area with others previously reported if this is the case.

Author Response

Dear editors and reviewers:

Thank you for giving us the opportunity to submit a revised draft of the manuscript “Spatial distribution, potential sources, and health risk of polycyclic aromatic hydrocarbons (PAHs) in the surface soils of Shanxi Province, north China” (ijerph-1878341) for publication in the Journal of Environmental Indicator. We appreciate the time and effort that you and the reviewers dedicated to providing feedback on our manuscript and are grateful for the insightful comments on and valuable improvements to our paper. We have incorporated most of the suggestions made by the reviewers. Those changes are highlighted in the manuscript. Please see below, in blue, for a point-by-point response to the reviewers’ comments and concerns. All page numbers refer to the revised manuscript file with tracked changes.

Replies to the reviewers’ comments:

Reviewer #1:

  1. Section 2.3. According to the procedure, the extracts were concentrated to 1 mL under a stream of nitrogen, and then they were concentrated to 1-2 mL? It's not clear.

Response: Thanks for your comment. We are sorry for our mistake. We have modified it as “1 mL” in Line 130, Page 7.

“The eluent was solvent-changed and concentrated to 1 mL before analysis.”

  1. Line 190. Baf is not a studied PHA.

Response: Thanks for your comment. We are sorry for our mistake. We have modified it as “Bbf” in Line 205, Page 10.

“The major PAHs were identified as Phe, Bbf, and Fla, accounting for 13.64%, 13.45%, and 12.48% of the total PAHs, respectively.” The least contribution of PAHs was Acy (0.40%) followed by Ace (0.44%).”

  1. Line 191. “The least dominant PAHs were Flu (0.40%) and Nap (0.44%)”. Are you referring to frequency or contribution? It does not correspond to Figure 2a. The one that contributes the least is Acy followed by Ace.

Response: Thanks for your comment. We are sorry for our mistake. We have modified it in Line 206-207, Page 10.

“The least contribution of PAHs was Acy (0.40%) followed by Ace (0.44%).”

  1. Did the authors find similar studies on PAH removal? It would be interesting to compare the results with the main points / strategies for elimination or degradation of the area with others previously reported if this is the case.

Response: Thanks for your comment. We have some results about PAH migration and removal in high altitude areas. We will continue to study and publish these achievements.

Reviewer 2 Report

Dear Authors,

This paper aimed to determine the spatial distribution, sources of pollution, and health risks of the poly aromatic hydrocarbons (PAHs) in Shanxi Province in China. This is a good initiative regarding the pollution assessment of the area and can provide an insightful context of the pollution in the area concerned. Scientists and policymakers can use this information to control the area's pollution. 

However, my primary concern is the soil sampling methods (see more comments on the attached pdf). The authors mentioned in their title that they assess the PAHs in surface soil. So, why do they collect soil from 10-30 cm depth? Why not from the surface, i.e., 0-30 cm? Surface soil, i.e., top 15 cm, is the most microbiologically active soil used for crop growth and getting all the PAHs.

Moreover, the authors did not mention the soil sampling methods or tools used for soil sampling. Sus as, are those collected as core soil samples? Or just by an augur? More soil sampling procedures are needed.

Also, what was the land area? Are the soil samples representative enough for the spatial analysis? Also, from the map and writing, I can see the site has a diverse topography. Are those samples represented both low and high land? Is the topography of the sampling area covered (i.e., are those samples representing the top, middle and bottom part of a slope?

The soil sample location needed to be described in this paper. Also, more accurate information on the soil sample collection procedures and description of the area, including the topography and soil, is essential as these are the most important driving factor for the distribution of the PAHs.

Please find the attached pdf for detailed comments and suggestions.

Best regards,

Author Response

Dear editors and reviewers:

Thank you for giving us the opportunity to submit a revised draft of the manuscript “Spatial distribution, potential sources, and health risk of polycyclic aromatic hydrocarbons (PAHs) in the surface soils of Shanxi Province, north China” (ijerph-1878341) for publication in the Journal of Environmental Indicator. We appreciate the time and effort that you and the reviewers dedicated to providing feedback on our manuscript and are grateful for the insightful comments on and valuable improvements to our paper. We have incorporated most of the suggestions made by the reviewers. Those changes are highlighted in the manuscript. Please see below, in blue, for a point-by-point response to the reviewers’ comments and concerns. All page numbers refer to the revised manuscript file with tracked changes.

Reviewer #2:

  1. The author tried to assess PAHs under different land use/land cover, however, in the following line the author mentioned about the industrial areas which are bit confusing. I suggest to focus on one category. i.e., either the land use or the various industrial areas. Also, the author should mention/include the land use in their title. For example, the author can say .... sources and spatial distribution in surface soil under different land use from.... Moreover, in the objective 2, the authors wanted to identify the sources of PAHs under different land use/land cover. Please try to be insistence in writing to avoid any confusion to the readers.

Response: Thanks for your comment. We have modified the title to “Spatial distribution, potential sources, and health risk of polycyclic aromatic hydrocarbons (PAHs) in the surface soils under different land-use covers of Shanxi Province, north China”. “Industrial areas” in Introduction refers Shanxi Province dominated by heavy industry. We deleted the confusing description of “industrial areas” in Line 87-89, Page 5.

“In addition, research on the differences in the sources and distribution of soil PAHs under different land-use covers is limited. In order to overcome these limitations, the current study aims to:”

  1. It should be either of one. Or if the geographical region covers both then the authors should write in that way.

Response: Thanks for your comment. After consulting the data, we determined that Shanxi is a semi-arid area. We have modified it as “Shanxi is a typical semi-arid area with four distinct seasons” in Line 100-101, Page 5.

  1. The authors mentioned in their title that they are assessing the PAHs in surface soil. So, why do they collect soil from 10-30 cm depth? Why not from the surface i.e., 0-30 cm? Surface soil i.e., top 15 cm is the most microbiologically active soil and is used for crop growth and getting all the PAHs. Moreover, the authors did not mention about the soil sampling methods or tools used for soil sampling. Sus as, are those collected as core soil samples? or just by an augur? More soil sampling procedures are needed. Also, what was the land area? Are the soil samples representative enough for the spatial analysis?

Response: Thanks for your comment. In fact, this is our mistake. A total of 173 surface soil samples from Shanxi Province (156,700 km2) were collected. Each sample was collected in a 100 m2 square area. The soil samples representative enough for the spatial analysis. We have modified the descriptions in Line 111-116, Page 6.

“For each sampling site, 5 sub-samples (each 1 kg) were taken from the same area (at a depth of 0–20 cm in a 100 m2 square area) with a stainless-steel shovel, and then bulked together to form one composite sample. A quarter of the composite sample was collected and stored in a polyethylene bag. Stones and residual roots in the soil samples were removed and freeze-dried for 48 h. Soil samples were sieved into 70 mesh particles prior to analysis.”

  1. Are those samples represented both low and high land? Is the topography of the sampling area covered (i.e., are those samples representing the top, middle and bottom part of a slope? The soil sample location needed to be described in this paper. Also, more accurate information on the soil sample collection procedures and description of the area including the topography and soil is essential as these are the most important driving fact.

Response: Thanks for your comment. We recorded detailed altitude and geomorphology during sampling. The next study will discuss the impact of altitude geomorphology, soil attributes, and vegetation cover types on PAHs in more detail. The altitude and geomorphology data of the sampling location are listed in the supporting information.

  1. This should not be the part of the materials and methods. Why this methods are good or important is not necessary here. But how the methods applied is more important. Providing a solid reference on the method is good enough. The authors may like to include these in the introduction section or discuss shortly in their discussion part (only if necessary).

Response: Thanks for your comment. We have deleted inappropriate descriptions in Line 149-151, Page 7.

“Positive Matrix Factorization (PMF) calculations was conducted based on the US EPA (Environmental Protection Agency) PMF 5.0 (Aminiyan et al., 2021; Sei et al., 2021).”

  1. Why the authors given the examples under the multivariate analysis? The authors should mention the exact names and procedures re: how and what statistical techniques were used to analysis of data.

Response: Thanks for your comment. We have modified our inappropriate expression in Line 169-170, Page 8.

“Multivariate statistical techniques including the creation of histograms, stacked histograms, and pie charts) were run on SigmaPlot 14.0”

  1. “]” delete. I think the R should be referenced. Again, these two lines are not necessary here. Remember the title of this paragraph. The authors are intended to mention the statistical and geo statistical analysis done here but not their efficiency or suitability. Be specific. Again, this can be explained or included as a footnote under the table or figures not here. If the authors want to mention here explicitly then make it short.

Response: Thanks for your comment. We have deleted “[” and referenced the related materials about R. We have deleted redundant descriptions about “spatial autocorrelation” and replaced them with references in Line 169-173, Page 8.

“Multivariate statistical techniques including the creation of histograms, stacked histograms, and pie charts) were run on SigmaPlot 14.0 (Systat Software, USA). Variations between different groups were determined using ANOVA with a significant set of p < 0.01. Box pots were drawn with R (R 4.1.3, AT&T Bell Laboratories) (Shalabh, 2022).”

“Spatial autocorrelation (Global Moran’s Ⅰ, between -1 and 1) was employed to analyze the spatial characteristics of soil samples (Sugg et al., 2021).”

Shalabh. (2022). Univariate, bivariate and multivariate statistics using R: quantitative tools for data analysis and data science. Journal of the Royal Statistical Society Series A, 185, 736–737. https://doi.org/10.1016/10.1111/rssa.12791

Sugg, M.M., Spaulding, T.J., Lane, S.J., Runkle, J. D., Iyer, L.S. (2021). Mapping community-level determinants of Covid-19 transmission in nursing homes: a multi-scale approach. Science of The Total Environment. 752, 141946. https://doi.org/10.1016/j.scitotenv.2020.141946

  1. Start with The before 'Phe'. Because a sentence shouldn't be started with a abbreviation.

Response: Thanks for your comment. We have modified it as “The Phe” in Line 193, Page 9.

  1. The author should mention the unit in the table. Also, a table must be standalone, i.e., all the abbreviation must be described under each table or figure. The readers are not going to find the abbreviation in the text of this manuscript. In this way the reader will lots track and interest.

Response: Thanks for your comment. We have modified Table 1.

Table 1 Concentrations of polycyclic aromatic hydrocarbons (PAHs) in the soil samples from Shanxi Province (ng g1).

Compounds

Min

(ng g1)

Max

(ng g1)

Mean

(ng g1)

SD

(ng g1)

Compounds

Min

(ng g1)

Max

(ng g1)

Mean

(ng g1)

SD

(ng g1)

Nap

2.59

79.60

13.28

10.21

Bkfc

0.30

72.85

10.60

13.11

Acy

0.09

10.92

1.61

1.45

Bapc

0.21

89.79

11.38

15.69

Ace

0.21

167.11

3.76

13.56

Inpc

0.29

71.62

11.13

13.36

Flu

1.11

31.07

6.33

4.19

Daac

0.18

25.25

4.05

4.68

Phe

5.03

235.34

32.92

32.12

Bgp

0.98

111.86

14.58

17.00

Ant

0.10

22.99

3.45

3.43

2-ring PAHs

2.59

79.60

13.28

10.21

Fla

1.51

221.37

26.48

33.61

3-ring PAHs

8.26

305.16

48.07

42.49

Pyr

1.08

160.87

19.85

25.14

4-ring PAHs

4.77

601.63

81.22

101.89

Baac

0.73

110.43

14.25

19.54

5-ring PAHs

2.33

324.73

51.87

65.32

Chrc

1.20

148.79

20.63

25.07

6-ring PAHs

1.79

192.97

29.76

34.66

Bbfc

1.06

181.07

29.89

36.88

∑16 PAHs

22.12

1337.82

224.21

243.99

Note: c Carcinogenic PAHs; naphthalene (Nap); acenaphthylene (Acy); acenaphthene (Ace); fluorene (Flu); phenanthrene (Phe); anthracene (Ant); fluoranthene (Fla); pyrene (Pyr); benz[a]anthracene (Baa); chrysene (Chr); benzo[b]fluoranthene (Bbf); benzo[k]fluoranthene (Bkf); benzo[a]pyrene (Bap); indeno[1,2,3-cd]pyrene (Inp); dibenzo[ah]anthracene (Daa); and benzo[ghi]perylene (Bgp).

  1. Again, all the abbreviation need to be described below the figure to be stand alone.

Response: Response: Thanks for your comment. We have added the abbreviation in Figure 2.

Fig. 2. Proportion of individual PAHs (a) and PAHs with different ring numbers (b) in the soil samples. naphthalene (Nap); acenaphthylene (Acy); acenaphthene (Ace); fluorene (Flu); phenanthrene (Phe); anthracene (Ant); fluoranthene (Fla); pyrene (Pyr); benz[a]anthracene (Baa); chrysene (Chr); benzo[b]fluoranthene (Bbf); benzo[k]fluoranthene (Bkf); benzo[a]pyrene (Bap); indeno[1,2,3-cd]pyrene (Inp); dibenzo[ah]anthracene (Daa); and benzo[ghi]perylene (Bgp).”

  1. Either sue "Land use" or "Land cover".

Response: Thanks for your comment. We have unified the description of the full text as “land-use cover”.

  1. Better not to say identification. The author can use "source of contamination or pollution". The wording of this sub-section reads like a methods. Consider rephrasing like "Carcinogenic risk" or different.

Response: Thanks for your comment. We have modified the description as “Source of contamination” (3.4) and “Carcinogenic risk” (3.5).

Reviewer 3 Report

(1)The selection of soil samples is of great significance to the accuracy of the research results. Section 2.2 of the paper gives the sampling situation of samples from different sources. Please provide the basis for the sampling field and quantity.

(2)The author may consider selecting some of the pollutant distribution results obtained in the research in the paper, and further compare it with the actual soil situation to highlight the reliability of the research results.

(3)The significant figures in the paper are not standardized, please revise them.

Author Response

Dear editors and reviewers:

Thank you for giving us the opportunity to submit a revised draft of the manuscript “Spatial distribution, potential sources, and health risk of polycyclic aromatic hydrocarbons (PAHs) in the surface soils of Shanxi Province, north China” (ijerph-1878341) for publication in the Journal of Environmental Indicator. We appreciate the time and effort that you and the reviewers dedicated to providing feedback on our manuscript and are grateful for the insightful comments on and valuable improvements to our paper. We have incorporated most of the suggestions made by the reviewers. Those changes are highlighted in the manuscript. Please see below, in blue, for a point-by-point response to the reviewers’ comments and concerns. All page numbers refer to the revised manuscript file with tracked changes.

Reviewer #3:

  1. The selection of soil samples is of great significance to the accuracy of the research results. Section 2.2 of the paper gives the sampling situation of samples from different sources. Please provide the basis for the sampling field and quantity.

Response: Thanks for your comment. We have added the descriptions in Line 111-116, Page 6. We recorded detailed altitude and geomorphology during sampling. The next study will discuss the impact of altitude geomorphology, soil attributes, and vegetation cover types on PAHs in more detail. The altitude and geomorphology data of the sampling location are listed in the supporting information.

“For each sampling site, 5 sub-samples (each 1 kg) were taken from the same area (at a depth of 0–20 cm in a 100 m2 square area) with a stainless-steel shovel, and then bulked together to form one composite sample. A quarter of the composite sample was collected and stored in a polyethylene bag. Stones and residual roots in the soil samples were removed and freeze-dried for 48 h. Soil samples were sieved into 70 mesh particles prior to analysis.”

  1. The author may consider selecting some of the pollutant distribution results obtained in the research in the paper, and further compare it with the actual soil situation to highlight the reliability of the research results.

Response: Thanks for your comment. Your suggestion is very good. We have already carried out some relevant work to compare the soils PAHs of different properties and crops. The experiment and analysis have not been completed. We will continue to carry out relevant research in the future.

  1. The significant figures in the paper are not standardized, please revise them.

Response: Thanks for your comment. We have modified and unified the significant figures in the paper. These changes have been marked in yellow.

Round 2

Reviewer 2 Report

Dear Authors,

Thank you for addressing most of the comments. However, I wonder how the depth changed this time from 10-30 cm to 0-20 cm. Which one is accurate?

Also, only one point sampling in a 100 square kilometre is a very low resolution for this assessment. 

Author Response

We are so sorry for our mistakes.  We did make a mistake about depth and unit. We have repeatedly determined the depth (0–20 cm) and sampling range (100 m2 square area).

"For each sampling site, 5 sub-samples (each 1 kg) were taken from the same area (at a depth of 0–20 cm in a 100 m2 square area) with a stainless steel shovel, and then bulked together to form one composite sample."

For detailed support, please refer to the papers published by our team before.

Li RJ, Cheng MC, Cui Y, et al. (2020). Distribution of the Soil PAHs and Health Risk Influenced by Coal Usage Processes in Taiyuan City, Northern China. Int. J. Environ. Res. Public Health, 17, 6319; doi:10.3390/ijerph17176319